# Systems Biology Approaches for the Improvement of Oncolytic Virus-Based Immunotherapies

**DOI:** 10.3390/cancers15041297

**Published:** 2023-02-17

**Authors:** Lorella Tripodi, Emanuele Sasso, Sara Feola, Ludovica Coluccino, Maria Vitale, Guido Leoni, Barbara Szomolay, Lucio Pastore, Vincenzo Cerullo

**Affiliations:** 1Dipartimento di Medicina Molecolare e Biotecnologie Mediche, Università degli Studi di Napoli Federico II, 80138 Naples, Italy; 2CEINGE Biotecnologie Avanzate Franco Salvatore, 80131 Naples, Italy; 3Laboratory of Immunovirotherapy, Drug Research Program, Faculty of Pharmacy, University of Helsinki, 00100 Helsinki, Finland; 4Translational Immunology Research Program (TRIMM), University of Helsinki, 00100 Helsinki, Finland; 5iCAN Digital Precision Cancer Medicine Flagship, University of Helsinki, 00100 Helsinki, Finland; 6Nouscom Srl, via Castel Romano 100, 00128 Rome, Italy; 7Systems Immunity Research Institute, Cardiff University School of Medicine, Cardiff CF14 4YS, UK

**Keywords:** oncolytic viruses (OVs), synthetic biology, system biology, epitope prediction

## Abstract

**Simple Summary:**

This review provides a roadmap to develop safe and effective OV-based future therapeutics by using several novel synthetic and system biology strategies. Integration of system and synthetic biology can improve the genetic design of OVs backbone, maintaining enough virulence-associated genes. Furthermore, specific computation pipelines can refine target discovery in several types of cancer, identifying the MHC-I and II-restricted peptide repertoire recognized by T-cells and reinforcing anticancer immune responses. Using these versatile approaches could encourage the development of a next-generation of OVs-based therapies, overcoming current challenges such as on-target, off-tumor effects, and variable clinical responses. The efficacy and safety profile of the current OVs-based regimens could be further enhanced with additional long-lasting clinical effects.

**Abstract:**

Oncolytic virus (OV)-based immunotherapy is mainly dependent on establishing an efficient cell-mediated antitumor immunity. OV-mediated antitumor immunity elicits a renewed antitumor reactivity, stimulating a T-cell response against tumor-associated antigens (TAAs) and recruiting natural killer cells within the tumor microenvironment (TME). Despite the fact that OVs are unspecific cancer vaccine platforms, to further enhance antitumor immunity, it is crucial to identify the potentially immunogenic T-cell restricted TAAs, the main key orchestrators in evoking a specific and durable cytotoxic T-cell response. Today, innovative approaches derived from systems biology are exploited to improve target discovery in several types of cancer and to identify the MHC-I and II restricted peptide repertoire recognized by T-cells. Using specific computation pipelines, it is possible to select the best tumor peptide candidates that can be efficiently vectorized and delivered by numerous OV-based platforms, in order to reinforce anticancer immune responses. Beyond the identification of TAAs, system biology can also support the engineering of OVs with improved oncotropism to reduce toxicity and maintain a sufficient portion of the wild-type virus virulence. Finally, these technologies can also pave the way towards a more rational design of armed OVs where a transgene of interest can be delivered to TME to develop an intratumoral gene therapy to enhance specific immune stimuli.

## 1. Introduction

In the past decade, immunotherapy has demonstrated great clinical success thanks to the ability to overcome tumor-mediated immunosuppression. The main advantage of cancer immunotherapy over other therapies, such as chemotherapy and radiotherapy, is the generation of a cancer-specific response with long-lasting effects [1]. Despite showing great initial promise, on further development, cancer immune-therapeutics—such as antibodies and chimeric antigen receptor (CAR) T-cells—have often failed to achieve the expected therapeutic benefits in severe and highly immunosuppressive diseases. Intrinsic and extrinsic resistance factors have been reviewed in many recent publications but, in general, it can be assumed that cancer cells can escape immune surveillance and antitumor immunity by two main mechanisms: (i) tumor cells are recognized as self; (ii) immune response is disabled by systemic and local immune suppression. These main limitations of immunotherapy are related to lack of targetable tumor antigens and poor immunogenicity of these tumors [2]. Therapeutic efficacy of immunomodulators such as cytokines, chemokines, and immune checkpoint inhibitors (ICIs) is often compromised when tumors are immune-excluded or profoundly immunosuppressed [3]. For this reason, combinational approaches of several immunomodulators are frequently investigated. Unfortunately, despite additional clinical benefits, systemic treatment with multiple immunomodulators frequently causes severe side effects, often requiring treatment discontinuation [4]. This evidence highlights a compelling need to explore efficient approaches to reshape the immunological state in the tumor microenvironment (TME) to reduce side effects and improve cancer immunotherapy.

Oncolytic viruses (OVs) are able to infect and replicate specifically in cancer cells, causing the direct lysis of tumor cells [5]. OV-mediated cancer cell lysis releases pathogen-associated molecular patterns (PAMPs), damage-associated molecular patterns (DAMPs), cell debris and tumor-associated antigens (TAAs). The latter recruit and activate dendritic cells (DCs) with consequent stimulation of specific T-cells that trigger strong and durable response against the tumor itself. Thanks to their ability to modulate TME, OVs are considered a class of active immunotherapeutics able to not only stimulate an immune response against TAAs, but also actively recruit lymphocytes and other immune cells [2]. In fact, in this scenario, antitumoral response overlaps with anti-viral response triggered against viral antigens that restrict the viral replication and spread, leading to a decrease in therapeutic efficacy. It might seem like a negative reaction, but detection of OV infection by the immune system has undeniably beneficial aspects. In response to viral infection, the intracellular interferon response triggers a cascade of signaling events that culminate with the release of cytokines and DAMPs, including the activation of natural killer (NKs) cells and phagocytes. The migration of viral antigens to the lymph nodes by antigen-presenting cells (APCs) elicits robust adaptive cellular immune responses, activating specific T-cells. T-cells exert cytotoxic effects on virus-infected cancer cells [5]. The great advantage of OVs is related to their ability to produce active and long-lasting antitumoral immune responses locally and systemically. Indeed, successful OV therapy in patients relies on the balance between anti-viral immune response that eliminate the virus and OV-induced anti-tumoral immunity that recognize TAAs, and neoantigens released from the virus-infected tumor cells. In addition, OV therapeutic efficacy can be greatly enhanced by combining with ICIs or by expressing and locally releasing immune modulators, further reducing the immunosuppressive state of TME with reduced side effects compared to systemic administration of these molecules [6]. While most of the clinical trials assessing combinations of OVs with ICIs are still ongoing, early observations from phase II and III trials suggest a trend of therapeutic benefits with evidence of both objective response and decreased tumor recurrence [6,7,8,9]. Together with these data, other clinical results suggest the need to further improve OV-based vaccine platforms: indeed, combining T-VEC and pembrolizumab has failed to produce significant benefits compared to pembrolizumab monotherapy [10].

In addition, improving cancer-targeting specificity of OVs can be a critical addition to improve their safety and efficacy. Even though toxicity does not appear to be a main issue in clinical settings, improved tropism towards cancer cells can be of relevance to reduce hepatotoxicity and, eventually, lead to intravenous OV administration, thereby improving clinical applicability. OVs can achieve their targeting to cancer cells thanks to the overlapping function that many tumor suppressor genes have with antiviral functions. The frequent inactivation of tumor suppressor genes in cancer makes tumors naturally prone and susceptible to the viral infection. Wild-type viruses naturally evolved several virulence-associated genes able to counteract antiviral pathways and trigger a tug-of-war between host and pathogen [11,12]. In this race between host antiviral innate immunity and immune evasion strategies, innate and immune reactions are most often able to prevent systemic infection and death. Based on this concept, most of the currently investigated or approved (T-VEC, G47∆) OVs have been generated by isolating strains with natural adaptation to cancer cells or by genetic engineering of wild-type viruses, leaving normal cells unharmed. Thanks to this approach, OVs maintain their lytic potential in cancer cells and trigger antiviral immune response. Beyond these known mechanisms, some OVs are naturally oncotropic by exploiting the biochemical alteration of cancer cell metabolism or thanks to receptors interacting with tumor associated ligands.

A large number of both DNA and RNA viruses have been engineered as OVs. The most common OVs used in clinical trials are adenoviruses (Ads), herpes simplex 1 (HSV-1), reoviruses and poxviruses [13]. Among these OVs, Ads have been engineered by attenuating mutations into virulence associated proteins (similarly to the modification of 34.5 in HSV). Ads contain a double-stranded linear DNA genome of 36 kb and are attractive due to relatively low pathogenic risk, high genome stability, a wide range of tissue tropism, and large DNA loading capacity (up to 8.5-kb foreign DNA). In the Ad genome, the E1 region contains two genes (E1A and E1B) that encode proteins required for a productive lytic cycle, even though E1A expression alone is sufficient to initiate the viral replication in different cancer cells [5,14,15]. A simple method to improve the selectivity of oncolytic adenovirus (OAd) is using cancer-specific promoters to regulate the E1A expression. As described by Huang et al., a winning strategy is to combine multiple inputs to further improve the cancer-targeting specificity [16]. In the next paragraphs, we will discuss the main efforts stemming from synthetic and system biology applied to the design of OVs with reduced side effects and a more robust antitumor immune response.

### Synthetic Biology as a Tool to Optimize OV Design

Advanced bioengineering from synthetic and systems biology are useful to overcome current challenges in the development of effective OV-based cancer immunotherapies, including off-target effects and poor clinical responses. Synthetic biology is an emerging field that focuses on the synthesis of complex, biology-based (or inspired) systems, which display functions that do not exist in nature [17]. Synthetic biology is a powerful tool to create therapeutics; in particular, it focuses on the development of synthetic genetic vectors (i.e., DNA, RNA, viral vectors) that encode networks of genes and synthetic genetic circuits (SGCs) to perform novel functions [18]. In 2020, Leventhal et al., using synthetic biology techniques, engineered a non-pathogenic strain of E. Coli in order to activate STING in phagocytic APCs in tumors. Certain bacteria are an ideal vector for STING-activation in APCs, as they are actively phagocytosed and trigger complementary immune pathways through the stimulation of pattern-recognition receptors (PRRs), such as the Toll-like receptors (TLRs). A phase I clinical trial (NCT04167137) is currently recruiting patients with advanced or metastatic malignancies who will be treated with the engineered bacterial vector described [19]. Such engineered SGCs are now primed to be packaged into OVs and programmed to coordinate local tissue responses, with in vitro studies demonstrating enhanced tumor cell targeting and generation of antitumor immunity [19,20].

An alternative strategy developed by Huang et al. consists in the design of a genetic circuit for a programmable OAd-based immunotherapy [16] able to selectively replicate in hepatocellular carcinoma cells and release immune mediators in the TME. The system is based on a genetic circuit with interchangeable cancer-selective promoters, microRNA target sites to detect distinct miRNA profiles of cancer cells, and genes that encode immune mediators such as interleukin-2 (IL-2), granulocyte-macrophage colony-stimulating factor (GM-CSF), or single-chain variable fragments (scFvs) against either programmed death-1 (PD-1) or programmed death-ligand 1 (PD-L1). The complex design of the circuit achieved increased tumor-selective replication as well as an enhanced host immune response. OAds were packaged with the synthetic genetic circuits and injected intratumorally into immunocompetent mice who were xenografted with hepatocellular carcinoma cells. In vivo data reported that all mice treated twice in one week with OAds, containing SGCs expressing GM-CSF and scFvs against PD-1, eliminated tumors within approximately a month. These promising results of programmable OV-based immunotherapy in a validated preclinical model of hepatocellular carcinoma may enable a rapid translation to other cancers. To improve design and evaluation of genetic circuits, system biology approaches are relevant to obtain integrated data from high-content omics platforms—analyzed with computational and statistical methods—that improve the characterization of the observed biological phenomena. In the next paragraph, we will discuss system biology applications for the development of OV-based immunotherapies.

## 2. Applications of System Biology in Target Discovery to Design New OV-Based Cancer Vaccines

Twenty years ago, Hood et al. described system biology as the investigation of “biological systems by systematically perturbing them genetically, biologically or chemically, monitoring the gene, protein and informational pathway responses, integrating these data in order to create mathematical models that describe the structure of the system and its response to individual perturbations” [21]. At present, system biology has been applied to cancer immunotherapy for new target discovery and comprehensive mechanistic understanding of therapies with identification of biomarkers for patients’ stratification and prediction of clinical response [22].

To date, therapeutic targets in cancer immunotherapy can be classified into two categories. The first class of targets is expressed on the surface of immune cells and can be either activated (i.e., by cytokine or agonistic antibody) or repressed (ICIs) to enhance antitumoral immune response. The targets in the second class are mainly TAAs or tumor neoantigens (TNAs) and can be used as immunogens to elicit a specific antitumoral immune response [23]. With an emphasis on the discovery and selection of immunogenic targets that can be delivered by OVs, we focus below on how systems biology approaches are being used to discover such targets and then optimize the design of engineered OVs.

### 2.1. Ligandome Analysis for Therapeutic Cancer Approaches

Ligandome, or immunopeptidome, describes the repertoire of naturally presented peptides within the MHC (major histocompatibility complex) or HLA (human-leukocyte antigen) complex on the cellular surface of every mammalian cell. The methods developed and applied for investigating the ligandome are referred to as immunopetidomics [24]. Some of different techniques used in this emerging field are direct isolation of MHC-restricted epitopes (i.e., immunopurification, soluble HLA, acid stripping), tandem mass-spectrometry for the identification of isolated peptides, and bioinformatic tools for downstream selection and prioritization of candidates (NetMHCpan 4.1, Gibbs clustering) [23,25]. Following the discovery of CD8+ T-cells, that are able to recognize and kill cancer cells in an MHC-I antigen-restricted manner, the identification of epitopes recognized by CD8+ T-cells for therapeutic cancer applications became of utmost importance. Moreover, the breakthrough of antibodies targeting immune-checkpoint molecules, such as PD-1, its ligand PD-L1, and cytotoxic T-cell-associated antigen 4 (CTL-A4) [26], has led to a new and strong interest in discovering suitable targets for CD8+T-cells [27]. Indeed, ICIs release the brakes of the immune system, stimulating effector functions of specific antitumor CD8+T-cells within the TME [28]. To this end, therapeutic cancer vaccines designed for the artificial generation and/or stimulation of CD8+ T-cells are needed [27,29].

In this context, state-of-the-art immunopeptidome and mass spectrometric methodologies have gained momentum, as the knowledge of MHC-I restricted peptides paves the way to the rational design of vaccines for immunotherapeutic approaches, providing novel treatment options [30]. For instance, ligandome can be exploited to capture changes in the MHC-I repertoire dictated by altered metabolism in cancer cells compared to the non-malignant counterpart for targeted immune interventions as in Löffler et al. [31]. These investigators characterized patients’ MHC-I restricted peptide repertoire in CRC (colon rectal cancer) and corresponding non-malignant tissue. The authors discovered a subset of cancer-specific non-mutated peptides not expressed in normal cells that can be used in immunotherapeutic approaches [31], avoiding the “on-target, off-tumor” effect. In another study, a highly sensitive ligandome workflow was applied to the identification of TNAs in melanoma samples collected from 25 different patients [32]. The authors demonstrated the feasibility of ligandome analysis for the identification of TNAs in patients, and that selected MHC-I TNAs were able to activate T-cell response in PBMCs derived from a melanoma patient [32].

Interestingly, ligandome analysis can be used to investigate changes in the MHC-I-restricted peptide repertoire in cancer cells upon specific treatments. Marin et al. observed that induced senescence of cancer cells could be exploited to generate potent anti-cancer immune protection in pre-clinical tumor models. Indeed, induced senescence modulated the MHC-I-restricted peptides, causing the expression of novel peptide targets for CD8+ T-cells absent in the parental cells. Moreover, therapies based on OVs may shape neo-antitumor CD8+ T-cell responses, modulating the MHC-I response, as shown in Murphy et al. [33]: an oncolytic reovirus modulated the expression of novel CD8+ T- cell targets in a murine ovarian cancer model. Overall, ligandome analysis identified specific self-peptides able to activate CD8+T-cells response [34]. Moreover, ligandome analysis is an effective method for the identification of non-canonical MHC-I restricted targets able to mediate CD8+ T-cell response. As shown in Chong [35] and Laumont [36], non-coding genomic regions are a source of MHC-I-restricted peptides that elicit antitumor immune responses. However, the sole combination of sensitive and accurate mass spectrometry (MS)-based ligandome analysis and high-throughput techniques uncovered immunogenic MHC-I restricted peptide targets for further development of T -cell-based treatment.

Using ligandome analysis, as a tool to isolate MHC-I restricted peptides recognized by CD8+ T-cells, has been successfully integrated into a streamlined pipeline for the generation of personalized cancer vaccines. Upon isolation and selection of the most immunogenic peptide candidates expressed on malignant cells, the pipeline generated an efficient therapeutic oncolytic cancer vaccine. This approach was successfully applied for the treatment of murine models of colon cancer [37] triple-negative breast cancer (TNBC) [38], and in both cell lines and tumor biopsies from human mesothelioma [39]. These technologies allow the rapid development of personalized oncolytic immunotherapies. For example, Capasso et al. show a rapid approach to combining tumor-specific MHC-I restricted peptides with an OAd, on the basis that the latter display a negatively charged capsid (around −30 mV) with tumor-specific peptides (neopeptides) made positively charged by the addition of a poly-lysine chain. This complex, named PeptiCRAD (Peptide Coated Conditionally replicating Adenovirus), is stable and is able in vivo to trigger a peptide-specific immune response due to the “delivery” of the peptides directly to professional antigen-presenting cells (APCs) (Figure 1) [39,40,41,42,43].

This same technology has been applied also to enveloped viruses, PeptiENV. Specifically, Ylosmaki and co-authors have shown that replacing the poly-lysine chain with a lipid-friendly anchor, the same technology could be applied to enveloped viruses. In particular, they have shown enhanced efficacy and tumor-specific immunogenicity utilizing oncolytic vaccinia virus and herpes simplex virus decorated with tumor-specific peptides [44]. Finally, the same authors have also applied these technologies to bacteria such as Bacillus Gulmette Guerin (BCG) [45] and FDA-approved vaccines, for example MMR [46] (Figure 2).

Ligandome analysis has been used to investigate MHC-I-restricted peptides upon proteolysis targeting chimera (PROTAC) treatment [47]. This later enhanced the proteolysis of a specific molecule, enhancing the MHC-I presentation of peptides derived from the primary PROTAC target and also from the PROTAC pathway, generating new immunogenic peptides for CD8+ T-cells [47]. The crosstalk between MHC-I-restricted peptides and CD8+T-cells plays a major role in cancer therapeutic approaches. As previously described, the knowledge of the MHC-I restricted peptides paves the way to expanding the efficacy of therapeutic cancer immunotherapies. Indeed, T-cell-based approaches could take advantage of the change in the MHC-I-restricted peptide repertoire.

In future applications, considering the rapid technological evolution of both immunopurification [48] and mass-spectrometry methodologies, the ligandome analysis is anticipated to become a tool accessible to a larger portion of researchers; this will allow a faster and more precise investigation in the changes of the MHC-I peptide repertoire upon different immunotherapeutic cancer treatments. Thus, ligandome analysis could guide T-cell-based therapies for a more tailored and personalized medicine.

### 2.2. Systems Biology Approaches for the Improvement of OV-Based Immunotherapies

Wild-type and genetically engineered OVs were designed to attack and kill cancer cells. Adenovirus is the most common OV in clinical trials and the most common tumor targets are melanoma and gastrointestinal cancers [13]. Nevertheless, to date, only four OVs have been approved for the treatment of cancers. In 2004, a genetically unmodified ECHO-7 strain enterovirus RIGVIR was approved for the treatment of melanoma in Latvia [49]. In 2005, a genetically modified adenovirus, H101, achieved approval for the treatment of nasopharyngeal carcinoma as a part of combination chemotherapy in China [50]. In 2015, the U.S. Food and Drug Administration approved T-VEC, an attenuated herpes simplex virus, for the treatment of recurrent melanoma [51]. Finally, based on positive results gained in early phases clinical trials, the Japan Ministry of Health has granted conditional and time-limited approval to Teserpaturev (G47∆; Delytact) for the treatment of malignant glioma. A PubMed search identified 157 results of clinical trials from 2001 to 2022, supporting the importance of OVs in cancer treatment. The search was conducted on 28 November 2022, using the keyword “oncolytic virus”, filtered for clinical trials and randomized clinical trials. To reach full therapeutic potential of OVs in the clinic, there are certain aspects of interactions between OVs and tumor cells that can be targeted for enhanced therapeutics: more specifically, improving oncolysis, enhancing OV efficacy by overcoming delivery limitations such as physical barriers, reducing tumor escape of the antiviral immune response, and modulating the immunosuppressive TME can all improve OV efficacy. Despite OVs being a promising therapeutic option for cancer, there is an urgent need to develop a standardized systems biology pipeline for OV immunotherapies. Below, we review the main computational approaches, summarized in Figure 3, for the generation of oncolytic cancer vaccines.

To minimize off-target effects of the OV therapy, advanced bioengineering technologies using principles of both synthetic and systems biology are needed. As previously described, synthetic biology aims to develop SGCs that combine DNA, protein, RNA components and demonstrate a range of cellular functions such as bi-stability [54], oscillation [55] and feedback [56]. Systems biology complements synthetic biology by enabling design and analysis of sophisticated genetic circuits. Synthetic and systems biology principles have been applied to develop programmable OV-based immunotherapies for the treatment of glioblastoma multiforme (GBM) [18]. SCGs are used to program OVs to coordinate the selectivity of viruses to tumor cells and local immune responses to GBM, on the basis of three major design principles.

First, promoters drive the expression of the SCG to provide protection against off-target effects of the OV therapy, such as radiosensitive promoters like Survivin [57]. Second, binding sites are used for differentially expressed miRNAs to enhance selectivity to tumor cells such as miR-21, miR-93, miR-196, miR-335, miR-7, miR-34a, miR-124a [52,58,59]. Third, pro-apoptotic and pro-inflammatory therapeutic payloads increased the local response to the OV therapy, such as pro-apoptopic genes such as sTRAIL and hBAX [60,61] and pro-inflammatory genes such as IL-2, GM-CSF, and scFvs against PD-1 and PD-L1 [16]. Some considerations are required to design programmable OV-based immunotherapies such as optimizing genetic circuit size [16,62], applying orthogonality principles [53] and mitigating transcriptional noise [63] to reduce off-target effects. Although genetic engineering techniques allow for optimizing OV delivery, specificity for tumor cells and payload capacity, the combination with appropriate computational strategies could make OVs safer and more potent for cancer immunotherapy. Some of these strategies are discussed below.

Anticancer and tumor-homing peptides have been reported as new therapeutic agents to treat cancer due to their advantages of being easily synthesizable, having high specificity and low toxicity [64]. Computational methods such as TumorHPD, AntiCP, ACPred, MLACP, ACPP, and ACPred-FL (Table 2 in [65]) can be used to design anticancer and tumor-homing peptides for improved OV therapy. It has been suggested that tumor-homing peptides with enhanced cell-penetrating capabilities could improve the targeted delivery of OVs. An approach might be to predict tumor-penetrating peptide (TPP) sequences and to optimize their aqueous solubility using bioinformatics tools such as CellPPD, SkipCPP-Pred, CPPpred, KELM-CPPpred, CPPred-RF, ccSOL and PROS (Table 2 in [65]).

TNAs, arising from mutations in cancer DNA, are becoming an important therapeutic modality in precision oncology [66]. Selective tumor destruction by OVs leads to neoantigen-specific T-cell responses, making OVs ideal companions for checkpoint blockade therapy [67]. A major advantage of this approach is that it drives immune responses against an array of TAAs and TNAs in a given patient/tumor, without defining them a priori and hence, it is considered “antigen-agnostic” [68]. Over 27 bioinformatics pipelines are available for neoantigen prediction, such as Vaxrank, ProTECT, Epidisco, pVACtools and NeoPredPipe (Table 1 in [69]).

Despite neoantigens being a promising target for immunotherapy, there are many technical challenges that arise due to tumor immune escape and weak immunogenicity, i.e., the ability of MHC-bound peptides to induce adaptive immunity [69]. A recent study has identified two key components of tumor antigen immunogenicity [70]. The first component involves “antigen presentation features” associated with HLA binding affinity, expression of the originating gene, dwell time of the HLA-peptide interaction and peptide hydrophobicity. The second component encompasses recognition features associated with “agretopicity” [71], i.e., the ratio of mutant binding affinity to wild-type binding affinity and “foreignness”, i.e., the probability of TCR recognition as inferred by the homology of the neoantigen to known T-cell antigen from the IEDB [72].

Mutation-driven drug resistance represents a challenge for precision oncology. The “one target, one ligand” concept does not provide a complete solution in cancer due to the multifactorial nature of the disease. The current cancer drug discovery research is shifting towards a new approach that simultaneously modulates more than one target. It has been shown that OVs can efficiently inhibit cancers displaying a multiple drug resistant phenotype [73], making them attractive therapeutic carriers from a multi-target pharmacology viewpoint. Genetic engineering techniques enable researchers to develop multifunctional OVs that act as multi-target ligands and provide a way to overcome tumor resistance. Computational tools can either predict the anticancer activity of chemical compounds, e.g., cancerIN and CDRUG, or, cytokine-inducing peptides, e.g., IL4Pred and IL17eScan (Table 2 in [65]).

As reported in the previous Section 2.1, there is increasing evidence that OVs, in addition to their cancer-killing characteristics, can induce the presentation of novel MHC class I tumor antigens and shape neo-antitumor CD8 T-cell responses [33]. In this regard, mass spectrometry (MS) methods for the identification and relative or absolute quantification of peptides in the immunopeptidome (“pMHC repertoire” or “ligandome”) have become increasingly popular [74]. Several immunopeptidomics-based pipelines, incorporating state-of-the-art MS, have been developed for personalized oncolytic immunotherapy [37,43]. These pipelines usually integrate ligandome analysis with OAd platforms, such as PeptiCRAd. The selection of the most optimal peptide candidates can be refined using different approaches, such as RNA-seq analysis and Homology Evaluation Xenopeptides (HEX) software [43]. The latter is a free tool developed in [75], and allows the identification of tumor antigens similar to viral antigens based on principles of molecular mimicry mediated by cross-reactive T-cells.

### 2.3. The Right One at the Right Time: Exploiting System Biology to Implement Safety and Efficacy of OVs

Recently, many companies and academic scientists have aimed to generate more potent OVs. Isolation of novel strains has been explored successfully by different groups, but their tumor stringency is still restricted by classical attenuating mutations. Moreover, safety of OVs remains a concern as emerging scientific literature underlines how the assumption of tumor selective replication could be questioned where splenic, liver and immune cells have been reported as infected by OVs. While infection of non-cancer cells could also be beneficial to alert the immune system, it is a double-edged sword with potential serious adverse effects. Particular attention is given to possible safety issues, such as cytokine storm, liver toxicity, environmental shedding, and reversion to wildtype [76,77]. More recently, alternative strategies to obtain tumor stringency have been explored. System biology and deep transcriptomic characterization of tumors allowed the generation of very extensive public data relating on tumor-restricted membrane-associated TAAs and tumor-specific promoters. The knowledge of membrane-associated TAAs, as in the case of CAR T-cells therapy, enabled researchers to engineer fully a virulent OV with restricted tropism to tumor cells by replacing viral glycoproteins involved in cell entry with antibody fragments targeting the TAAs of interest (e.g., HER2, PSMA, MSLN) [78,79].

As an alternative, to preserve lytic activity of OVs without being at the expense of their safety profile, transcriptomic tumor analysis can be exploited to identify tumor-restricted promoters able to drive the expression of virulence-associated genes. This is the case for the conditionally replicating viruses implemented into several viral platforms (e.g., HSV-1, Ad). Beyond the classical tumor-restricted promoters (e.g., TERT, c-Myc), with the development of advanced sequencing technology, novel approaches of detecting genome-wide promoter activities are available. By this approach, Chiocca and colleagues at Dana-Farber Cancer Institute identified the nestin promoter to be tumor restricted, and developed rQNestin34.5v.2 (rQNestin) to treat brain tumors which is currently investigated in clinical trial NCT03152318 [80,81,82,83,84,85]. Tropism and transcriptional restriction can also be combined to get a safer oncolytic viral platform [86]. More recently, system biology applied to non-coding genes (i.e., miRNA and lncRNAs) has also become an appealing field in oncovirotherapy to both improve safety and antitumor efficacy. The first has been applied by Mazzacurati et al., where the authors searched for Cancer dysregulated miRNAs, a class of small non-coding RNA molecules that has been recognized as an important player in tumorigenesis and the establishment of TME. Such dysregulation identifies miRNA signatures that can distinguish between normal cells and tumor cells in different malignancies and can provide selectivity of OVs [87]. They exploited miRNAs downregulated in cancer to restrict the translation of herpetic transcription factor ICP4 to tumors cells lacking a specific miRNA (i.e., miR124) by incorporating miRNA response elements (MREs) into the 3′ UTR of the corresponding alpha4 gene locus [59]. MREs can reduce OVs in healthy tissue and, hence, avoid undesired toxicity associated with viral tropism when administered systemically [88]. Computational tools such as MiRanda [89] and miRmap [90] can help to predict the MRE within the viral genome and their target site repression strength.

Due to technological and cultural restraints, lncRNAs have been neglected for years. The application of novel bioinformatic pipelines depicts a very complex regulation and way of action of these RNAs that is also crucial in cancer. The versatility of OVs along with the rapid progress of engineering technologies have paved the way to next-generation arming strategies where lncRNAs can also be exploited as miRNA sponge to sequester oncomiRs into a cancer cell [91,92]. On the subject of arming strategies, OVs have been used for years as intratumoral gene-therapy carriers to express transgenes of interest. GM-CSF was the most common transgene, included into 25% of OVs in clinical trial and into the T-VEC thanks to its broad immune effect [13,93,94]. Despite preclinical data comparing unarmed vs GM-CSF armed viruses demonstrating benefits on survival, the clinical relevance is still debated [95]. To date, it has been demonstrated that different immunomodulatory genes can exert a more significant antitumor response compared to GM-CSF. In particular, by exploiting local tumor delivery of OV-encoded transgenes, we are assisting in the renaissance of signaling molecules with well-known adverse effects when administered systemically as proteins. This is the case of IL-2 and IL-12 that can be safely delivered to tumors as a payload into OVs [96,97,98,99].

While the encoding of immunomodulators such as cytokines and costimulatory molecules, is an obvious approach, the current available technologies of transcriptome profiling offer the opportunity to monitor the immunological perturbation induced by OVs themselves and to encode payloads aimed to push on the right molecular switches of the immune system at the right time. The upregulation of costimulatory receptors in tumor infiltrating lymphocytes is one of the milestones of oncovirotherapy. While the upregulation of costimulatory molecules has been reported in many scientific papers, often, their counterpart ligand does not appear to be accordingly upregulated, resulting in sub-optimal immune activation [100]. Zamarin and colleagues described how the upregulation of inducible costimulator (ICOS) in Newcastle disease virus (NDV)-treated tumors was not coupled to the presence of the matched inducible costimulatory-ligand (ICOSL). To get the best benefit of NDV-based oncovirotherapy, they trans-complemented the ICOS function by encoding ICOSL into the NDV vector, thus demonstrating a synergistic antitumor effect with CTLA-4 blockade [101]. This intuitive approach can be broadly applied to oncolytic virotherapy to complement the function of cytokine or chemokine receptors by encoding the matching payload into OVs. Transcriptomic profiling can also reveal profound immunosuppressive signatures where single payloads could not be enough to unlock the antitumor immune response. Influx of antitumor T-cells into the tumor could indeed be suboptimal due to metabolic insufficiencies or due to the presence of broadly acting immunosuppressive stimuli (e.g., adenosine). The profiling of tumor infiltrate of aggressive melanomas, assessed in Rivadeneira et al., revealed leptin as a key adipokine to release an immune brake into the tumor. The ectopic expression of leptin by oncolytic vaccinia virus reinforced the antitumor immune response by reprogramming T-cell fate [102].

Similarly, the immunosuppressive adenosine purinergic pathway has been reported as relevant in many cancer indications, with a rising number of novel clinical trials over the last ten years. Despite inhibiting the ADO pathway with systemically administered therapeutics (e.g., CD39 and CD73 inhibitors) is of interest, tumor-restricted targeting of ADO could reduce side effects and improve antitumor benefits. A signal peptide embedded adenosine deaminase has recently been generated and encoded into an HER2 targeted oncolytic herpes virus to enhance clearance of tumor localized adenosine [103,104]. Finally, more recently, the single cell RNA-seq (scRNA-seq) perspective of OV-treated cancer cells has now also been uncovered. This application demonstrated how some type of cells, previously thought of as not proficient for viral replication, can sustain T-VEC infection, and simultaneously, transgene expression. Knowing where and when a transgene can be encoded can sustain the concept of encoding “the right one at the right time” to further improve oncolytic virotherapy [105]. These results demonstrate how the era of system biology can revolutionize the field of oncovirotherapy as a powerful novel approach to directly target cancer specific pathways with the aim of improving safety and rationally enhancing antitumor immune responses (Figure 4).

### 2.4. Other Bioinformatic Tools to Find Out Immunogenic Peptides/Neoantigens to Insert in Vectors

Tumor cells accumulate multiple genomic alterations compared to healthy cells. The vast majority of alterations are substitutions or insertion/deletion of single or multiple DNA nucleotides; however, many tumors frequently present structural rearrangements, gene fusions and other events such as tumor-specific alternative RNA splicing. When the alterations happen in expressed sequences, neopeptides are produced. The advent of Next-Generation Sequencing (NGS) highlighted that the total number of somatic mutations and corresponding neopeptides vary considerably with tumor histology and from patient to patient. Among the hundreds of neopeptides detected in patients with tumors with high mutational burden such as melanoma and non-small-cell lung cancer (NSCLC), only a very limited fraction of them (1–2%) has the potential to stimulate the immune system to elicit a response against tumor cells and is defined as TNAs [106]. There is compelling evidence that supports the concept that TNAs represent a promising target for vaccination when combined with ICIs-based immunotherapy [107,108,109]. Indeed, TNAs have been shown to play a significant role in recognition and killing of tumor cells by CD8 and CD4 T-cell mediated immune responses [110]. Moreover, they are safe because they are present exclusively in tumor cells and not in normal cells, thereby reducing the risk of inducing auto-immune reactions. In addition, they are able to trigger immune response against other TAAs [111].

There are two major limitations in the selection of TNA targets that can be delivered by OVs. The first is that for the majority of tumors, very few somatic mutations are shared among patients. Therefore, the design of a candidate vaccine requires a personalized analysis of each patient tumor [112]. The second and most important aspect is that almost all the currently available cancer vaccine platforms, including OVs, have limited capacity to deliver a large number of peptides, making it impossible to target all or even most of the tumor-specific neopeptides, thus raising the need for efficient methods to select the neopeptides that have the highest likelihood of being neoantigens [70]. The fundamental requirements for being an effective TNA are that a neopeptide is abundant, efficiently presented by MHC class I and II complexes and recognized by the T-cell receptors (Figure 5).

Nowadays, neopeptides abundance and the likelihood of binding to MHC can be determined with very good accuracy. Developing accurate tools for the detection of somatic mutations encoding neopeptides, and then potential TNAs, was made possible by the availability of big datasets of tumor biopsies characterized by NGS; the availability of novel tumor proteogenomic MS spectrometry data; and the implementation of machine learning-based predictors that can estimate the likelihood of the binding of detected neopeptides to MHC-I and II,. On the other hand, prediction that a TNA is recognized by a T-cell is still a challenging task due to the complexity of physical mechanisms that influence how TCRs bind peptides in complex with the MHC protein [113]. Currently available TNA prioritization methods can be categorized into three groups according to the way in which the three previously mentioned parameters are used to rank the lists of possible TNAs.

Sequential filtering methods apply several user-specified filters in order to select candidate TNAs that satisfy all the specified criteria. A representative tool in this category is pVACseq and its recent implemented version, pVACtools [114]. Supported filters include variant expression, variant allelic fraction (VAF), and likelihood of neopeptide binding affinity using MHC class I and class II predictions. Moreover, the tool also includes the possibility of predicting stability of MHC-neopeptide complexes by using the software NetMHCStabPan [115]. Similarly, TIMiner software utilizes a similar approach to filter neopeptides that are encoded by expressed genes. The added value provided by TIminer is the possibility of further analyzing RNA-seq data to derive an estimate of the tumor-infiltrating immune cells detected in the analyzed biopsy, thus providing an estimate of the immunogenicity of the tumor [116]. An interesting method is ProGeo-Neo that incorporates a module to compare the neopeptides identified by analyzing genomic data with mass spectrometry profiles collected from the same tumor [117]. An additional filter can be applied to select neopeptides derived from SNVs that have higher similarity to microbial peptides, and therefore have a better chance of being recognized as non-self. The major drawback of the tool is that the complete workflow requires data from genomic and proteomic platforms, and this is not optimal in a clinical scenario in which, from the time of collection of the biopsy to the production of the vaccine, it is necessary to run as fast as possible in order to avoid the tumor changing due to immunoediting events.

Multiparametric score ranking methods implement scoring functions that combine different parameters to create a ranked list of potential neoepitopes arising from the input variants. The advantage with respect to the sequential filtering is that, with these methods, it is possible to rank all the neopeptides detected in a tumor biopsy by also including those that are good only for some of the required filtering criteria. A representative software of this category is MuPeXi, that utilizes a multiplicative ranking function to elaborate a score which relies on the expression of the gene encoding the mutant peptide, the allele frequency of the mutation, and the likelihood that the patient’s MHC recognizes only the mutated and not the wild-type peptide [118]. The tool has been validated by estimating its capability to experimentally discriminate validated TNAs among lists of candidate neopeptides, resulting in a significantly improved performance compared to pVACSeq. The major drawback of the MuPeXi pipeline is that the ranking is applied on the level of minimal peptide bound to MHC rather than the variants. Therefore, the user has to decide on an approach to aggregate the results in cases where multiple neopeptides resulting from one variant are interleaved with neopeptides from a different variant in the ranked list.

The expression levels of neopeptides is an important feature, so that it is necessary to further improving the performance of neoantigen prediction algorithms [119]. Tools like VaxRank and VENUS incorporate metrics in their score that account for variant-specific expression based on the supporting RNA-Seq read counts determined in tumor biopsy [120]. VENUS additionally combines the abundance of the transcripts carrying the mutation with the allele frequency of the mutation and the likelihood of MHC-I binding of the encoded neopeptide in order to produce a final rank that better reflects the degree of neopeptide presentation to the immune system [121]. Epi-Seq instead relies exclusively on RNA-seq data by performing the calling of tumor-specific variants that encode neopeptides directly on RNA tumor biopsy [122]. This tool can be particularly useful when only RNA-seq tumor data are available; however, the lists of detected variants encoding neopeptides need to be carefully examined in order to avoid the inclusion of peptides derived from non-annotated germline variants.

Methods based on the prediction of the interaction among TCRs and neopeptide-MHC complexes are still at the exploratory stage. The principal factor that limits the development of better predictors is intrinsic to the necessity for TCRs to be “plastic” to bind to numerous potential antigens, with high affinity and with an order of magnitude larger than the number of unique TCRs in an individual. A combination of experimental and computational approaches has shown that a single TCR can recognize more than a million different peptides [123]. TCR cross-reactivity is fundamental to the immune system and is driven by the fact that TCRs focus on “hot-spot” regions of the peptide that can be structurally and chemically similar between different agonist ligands. Outside of the hot-spot, more sequence diversity is permitted [113]. Currently available predictors rely on experimental datasets of limited size to extract rules that determine the characteristics of the hot-spots on the surface of peptides bound to MHCs. A representative method of this group is the neoantigen quality model that has been used to characterize the extent of immunoediting in long term survivor pancreatic cancers [124]. The neoantigen quality model is based on the concept that a neoantigen will be immunogenic if there is a TCR able to discriminate it than its wild-type counterpart. The immunogenicity estimate is determined by two features: the differential MHC presentation, determined as the ratio of the predicted MHC-binding affinity between mutant and wild-type neopeptide, and the differential T-cell reactivity. The latter is determined by evaluating whether the different amino acid(s) could lead to an improvement in the strength of the binding with TCR, estimated using an experimental affinity dataset of one TCR complexed with 1026 mutant peptides of the same model antigen. On the other hand, PRIME software utilizes a logistic regression model that learned on a dataset of 31,896 ligands naturally presented by 12 HLA-I [125]. For every allele, the authors determined a frequency substitution matrix of specific positions between the fourth (P4) and the second to last amino acid of the MHC-bound peptide. The amino acid preferences at these positions are generally much less influenced by the binding to HLA-I, thereby preventing an important confounding factor when analyzing TCR binding and recognition. They then combined the predicted binding to HLA-I with the frequency of each amino acid at these positions, and used this matrix to train their predictor. PRIME probability scores are correlated with monomeric pHLA-TCR dissociation kinetics even though the reported correlation is lower than the one observed on the same dataset by testing a measure of dissimilarity of the neopeptide to self-proteome.

All the mentioned algorithms were developed independently by different teams and evaluated on datasets from patients with different tumor types. This heterogeneity prevents a direct comparison of the performances of each method. To address this limit, the Tumor Neoantigen Selection Alliance (TESLA) has recently been created, which is a global community-based initiative with the principal aim of benchmarking neoantigen prediction methods and characterizing the immunogenicity of TNAs selected by analyzing the mutanome of the same tumor patients [70]. The first contest among 25 TNA prediction algorithms was completed in 2020. All the teams analyzed the same unbiased dataset of NGS data from three melanoma and three NSCLC patients. The best ranked TNAs in each patient (median of 51 neoantigens per patient) were tested for immunogenicity by pMHC multimer-based assays. In according to assay results, 6% of tested TNAs were found to be immunogenic (37/608; median of three TNAs per patient). Four peptide features resulted in significantly enriched immunogenic TNAs, namely, the strong predicted MHC binding, the high abundance of the TNAs in tumors, the binding stability of the MHC-I/peptide complex, and the mean hydrophobicity of MHC-bound peptides. Predictions made from the different teams were poorly overlapping. Only 20% of neoantigens were selected by more than one team; in addition, none of the teams included more than 20 of the 37 immunogenic peptides in their top 100 predictions.

## 3. Conclusions

A great deal of progress has been made in addressing some of the challenges in the OV translation to clinical practice. Genetic engineering strategies have helped to enhance the safety and efficacy of OVs, and several studies have shown that OVs and other existing cancer therapies, such as immunomodulators and ICIs, can act in synergy to optimize therapeutic efficacy and to overcome potential resistance against either individual treatment modality. Despite all the progress, however, there is still a long way to go and space for improvement. In this review, we have summarized the various applications of synthetic and systems biology for novel, rationally-designed OV-based therapeutics that can obtain additional clinical benefits.

In particular, the great potential use of these applications is to overcome the current challenges faced in developing safe and effective OV-based immunotherapies. The SGCs packaged into OAds showed enhanced tumor cell targeting and the generation of potent antitumor immunity, in vitro and in vivo. To date, ligandome analysis has been successfully integrated into a streamlined pipeline for the generation of personalized cancer vaccines in a murine model of colon cancer, of TNBC, and in human mesothelioma. The selection of the best peptide candidates to insert in the backbone of OVs can be refined using a free tool, HEX, able to identify tumor antigens similar to viral and bacterial ones based on molecular mimicry mediated by cross-reactive T-cells. To implement safety and efficacy of OVs, the technologies of transciptome profiling allow the creation of a big library of TAAs and tumor-specific promoters, useful for modifying the OVs with restricted tropism to tumor cells. Another appealing application is to manage the extensive immune network triggered by OVs, encoding payloads aimed at turning on the right switches of the immune system at the right time. Finally, numerous bioinformatic tools are available to find out immunogenic neoantigens and target peptides that can be inserted in oncolytic viral vectors. We retain that the synthetic and system biology strategies represent powerful tools to design a new category of OV-based immunotherapy, that can revolutionize the way cancer is treated. We hope that the principles outlined here will help guide the additional preclinical and clinical research development required to contribute to precision oncology research.

## Figures and Tables

**Figure 1 cancers-15-01297-f001:**
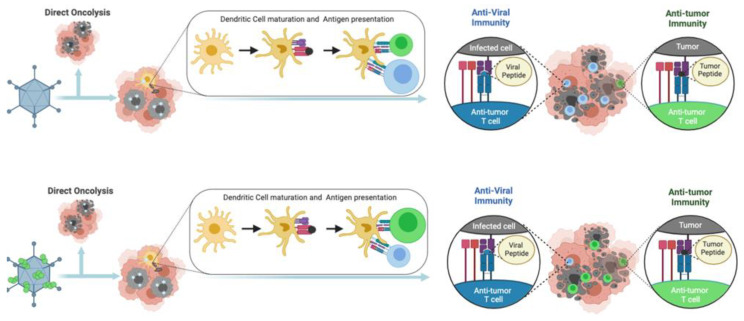
Schematic of the mode of action of PeptiCRAd.

**Figure 2 cancers-15-01297-f002:**
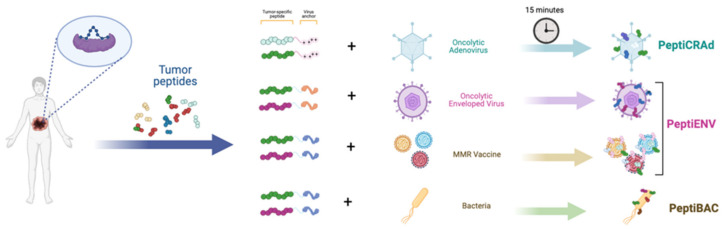
Schematic representing the Pepti-technology used to the rapid development of personalizing oncolytic immunotherapy platforms.

**Figure 3 cancers-15-01297-f003:**
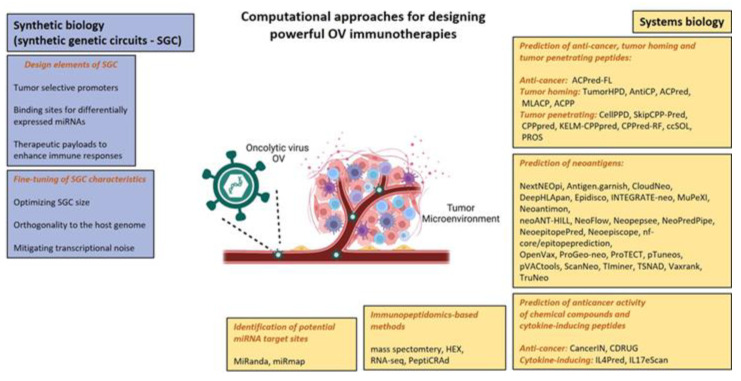
Summary of main computational approaches used in designing effective oncolytic virus immunotherapies, based on [18,37,52,53]. Adapted from “Tumor microenvironment”, by BioRender.com (2022). Retrieved from https://app.biorender.com/biorender-templates (accessed on 17 November 2022).

**Figure 4 cancers-15-01297-f004:**
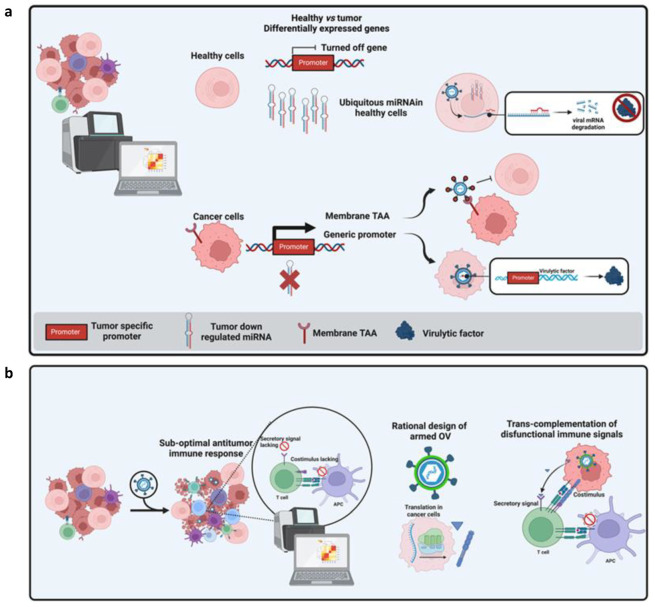
(**a**) Modalities to restrict tropism or replication of OVs based on multi-omics data. In the cartoon, a tumor transcriptome is compared to those of healthy tissues to identify differentially expressed genes (DEG). These genes have been reported as down-regulated or up-regulated in tumors. Different approaches can exploit these DEGs: when a miRNA is solely expressed in healthy tissues but is absent in cancer cells, matching seed sequences can be ectopically incorporated into the 3′ UTR of the viral gene to inhibit the translation of a given virulence factor in healthy tissues while preserving its functionality in tumor cells (1). By contrast, when a tumor-associated promoter is identified, its regulatory region can be used to replace endogenous viral promoters to allow transcription only in tumor cells (3). Finally, a membrane associated TAA, represented as generic receptor in the cartoon, can be targeted by targeted OVs where viral glycoproteins can host a bait to interact with the TAA. This bait can be an antibody fragment or even a ligand of TAA. (**b**) The concept of rational arming of OVs is represented. Perturbation in transcriptomic profiling of tumor bulk make it possible to identify sub-optimally activated immune pathways whose function can be improved by encoding the matching factors into the OV genome.

**Figure 5 cancers-15-01297-f005:**
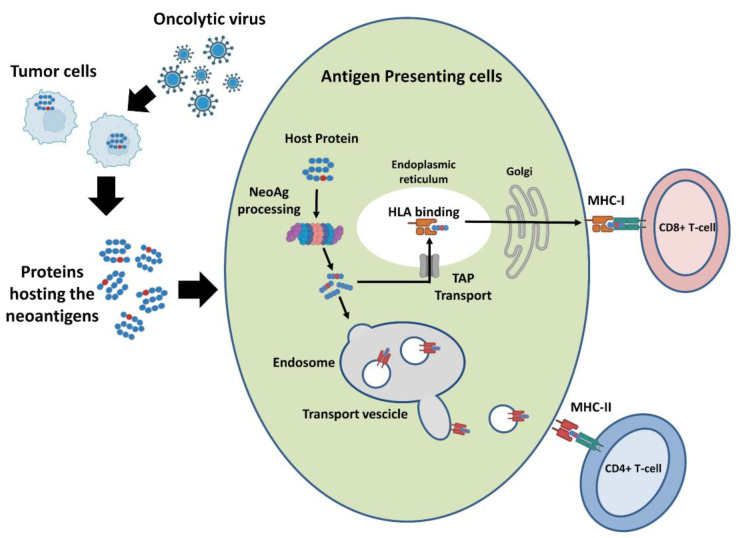
Pathway of TNAs presentation. Tumor cells encode a variable number of neopeptides with several abundance levels. The two major classes of proteins specialized for antigen presentation are the MHC class I and class II molecules, which present antigenic peptides to CD8+ T-cells and CD4+ T-cells, respectively. Only a small part of the expressed neopeptides is bound to MHCs and a lower fraction of these is recognized by T-cells.

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
