# Peer review of "Systems Biology Approaches for the Improvement of Oncolytic Virus-Based Immunotherapies"

_cancers, 2023, doi:10.3390/cancers15041297_

Round 1

Reviewer 1 Report

The review by Tripodi et al is contemporary and well structured. The reading is interesting and enlightening. I have only a few comments:

1) The authors refer to the aim of reducing toxicity of OVs at some sections. What type of toxicity noted in patients do they refer to? This is not very clear and then not well describe if the method of reducing toxicity is relevant in the clinical setting?

2) In terms of T cell mediated responses, the authors focus on anti-tumor T cell responses but what about the anti-adenoviral T cell responses and the likely contribution of such T cells to kill OV infected tumor cells. Is this beneficial or problematic in terms of hopefully gained anti-tumor T cell responses? 

3) The authors claim that oncolytic viruses induce long lasting antitumoral immune responses even without TAA-specific stimulation of the immune system. As the tumor cell will provide all relevant TAAs how do the authors mean that TAA-specific stimulation is not occurring and how could T cells be generated for long term immunity without it?

4) Please provide the commercial conflicts of interest.

Author Response

1) The authors refer to the aim of reducing toxicity of OVs at some sections. What type of toxicity noted in patients do they refer to? This is not very clear and then not well describe if the method of reducing toxicity is relevant in the clinical setting?

Response: Intravenous administration is easier and more accessible. It might also be more effective in treating metastasis; however, it could definitively pose additional health risks for patients and its environment, in particular, resulting in liver toxicity and viral shedding. Indeed, intratumoral route of administration increases the local viral dose in the tumor and hence obtain improved oncolytic efficacy; however, it presents several limitations for tumors located in sites difficult to reach. For these reasons, focusing on new strategies aimed at reducing systemic toxicity of OVs and minimizing side effects, can drastically improve clinical outcomes. We reviewed the text to add this information:

Line 103-106: “Even though toxicity does not appear to be a main issue in clinical settings, improved tropism towards cancer cells can be of relevance to reduce hepatotoxicity and, eventually, lead to intravenous OV administration, therefore improving clinical applicability”

2) In terms of T cell mediated responses, the authors focus on anti-tumor T cell responses but what about the anti-adenoviral T cell responses and the likely contribution of such T cells to kill OV infected tumor cells. Is this beneficial or problematic in terms of hopefully gained antitumor T cell responses?

Response: Anti-adenoviral T-cell response is relevant to recruit lymphocytes, dendritic cells (DCs) and other immune cells in the tumor microenvironment. OVs infect and lyse preferentially cancer cells, triggering an immunogenic cell death (ICD) with the consequential releasing of cell debris, viral antigens, damage-associated molecular patterns (DAMPs), and pathogen-associated molecular patterns (PAMPs). The latter mediate the first danger signaling and recruit specific T-cells to the tumor site, directed to viral progeny. In addition, OVs infection also mediate the activation of natural killer (NKs) cells which in turn contribute to the inflammatory tumor milieu. Simultaneously, oncolysis causes release of tumor-associated antigens (TAAs), which will recruit DCs and promote their maturation. So, mature DCs present TAAs to local and distant T-cells. In this way, the primary anti-adenoviral immune response is useful to create a pro-inflammatory environment that contributes and boost specific anti-tumor responses.

3) The authors claim that oncolytic viruses induce long lasting antitumoral immune responses even without TAA-specific stimulation of the immune system. As the tumor cell will provide all relevant TAAs how do the authors mean that TAA-specific stimulation is not occurring and how could T cells be generated for long term immunity without it?

Response: As described in the previous answers, OVs tumor infection stimulates the immune system inducing not only antiviral responses but also systemic and long-lasting anti-cancer responses. We reviewed the text to better explain and clarify the immune responses triggered by OVs infection, as suggested in comments 2 and 3:

Line 63-84: “The OVs-mediated cancer cell lysis releases pathogen-associated molecular patterns (PAMPs), damage-associated molecular patterns (DAMPs), cell debris and tumor-associated antigens (TAAs). The latter recruit and activate dendritic cells (DCs) with consequent stimulation of specific T -cells that trigger strong and durable response against the tumor itself. Thanks to their ability to modulate tumor microenvironment, OVs are considered a class of active immunotherapeutics able to not only stimulate an immune response against TAAs, but also actively recruit lymphocytes and other immune cells [2]. In fact, in this scenario antitumoral response overlaps with anti-viral response triggered against viral antigens that restrict the viral replication and spread, leading to a decrease in therapeutic efficacy. It might seem like a negative reaction but the detection of OV infection by the immune system has undeniably beneficial aspects. In response to viral infection, the intracellular interferon response triggers a cascade of signaling events that culminate with the release of cytokines and DAMPs, including the activation of natural killer (NKs) cells and phagocytes. The migration of viral antigens to the lymph nodes by antigen-presenting cells (APCs) elicits robust adaptive cellular immune responses, activating specific T-cells. T-cells exert cytotoxic effects on virus-infected cancer cells [5]. The great advantage of OVs is related to their ability to produce active and long-lasting antitumoral immune responses locally and systemically. Indeed, successful OVs therapy in patients relies on the balance between anti-viral immune response that eliminate the virus and OV-induced anti-tumoral immunity that recognize TAAs, and neoantigens released from the virus-infected tumor cells”.

4) Please provide the commercial conflicts of interest.

Response: We thank the reviewer for noticing it. We provided to add it in the revised version.

Reviewer 2 Report

The review manuscript is well-written, structured, analytical, and illustrated. The concept has been proved to work well in the immune competent animal models. However, it is still unclear how TAAs or TNAs, delivered by a virus into a tumor bed of a patient, will additionally/effectively promote an immune response if  severe local and systemic immune suppression is observed in the majority of patients.

English language and style are minor spell check required:

Line 52 “These main limitations of immunotherapy, are related

Line 81 “tumor suppressor genes in cancer, makes tumors”

Line 82 “susceptible to viral the infection”

Line 107 the reference[16] Huang et al. should appear in the text earlier before the last sentence of the paragraph.

Line 127 “in the design a genetic circuit”

Line 222 "Capasso et al., show a rapid approach"

Line 239 "already approved vaccines for example MMR"

and others...

Author Response

1) However, it is still unclear how TAAS or TNAS, delivered by a virus into a tumor bed of a patient, will additionally/effectively promote an immune response if severe local and systemic immune suppression is observed in the majority of patients.

Response: In several types of tumors, the process of immuno-editing favors an immunosuppressive tumor microenvironment characterized by a poor presence of tumor-infiltrating lymphocytes (TILs). OVs-tumor infection promotes a pro-inflammatory environment with the consequent recruitment of the immune cells such as lymphocytes, dendritic cells (DCs), natural killer cells (NKs), and release of cytokines, switching the tumor immuno-suppressive environment to a highly pro-inflammatory environment. In this scenario, tumor-associated antigens (TAAs) and tumor neo-antigens (TNAs) delivered by OVs promote the maturation of the recruited DCs. Mature dendritic cells present the delivered TAAs and TNAs to local and distant T-cells. In this way, we can induce powerful systemic and long-lasting anti-cancer immune responses.

2)English language and style are minor spell check required.

Response: We thank the reviewer for noticing it. We revised the English language and corrected the reported errors and the others.

Reviewer 3 Report

The authors present an extensive review on system biology strategies directed to improve safety and efficacy of oncolytic virotherapy, focused on immunotherapeutic aspects. The review is ample and touch many interesting development in the field.

Author Response

We thank you to appreciate our manuscript